# Selective toxicity of ascorbic acid and hydrogen peroxide on human tenon cells without harming scleral cells *in vitro*: A possible alternative to non-selective mitomycin C?

**Heiko Fuchs[1][*], Xiaonan Hu[1], Roland Meister[1], Yuqing Huang[1], Martin C. Bartram[1], Amelie Pielen[1,2], Bernd Junker[1,2], Jan Tode[1], Carsten Framme[1]**

1 Department of Ophthalmology, University Eye Hospital, Hannover Medical School, Hannover, Germany,
2 Maximilians-Augenklinik, Nürnberg, Germany

☯ These authors contributed equally to this work.
* fuchs.heiko@mh-hannover.de

## Abstract

### Background

Glaucoma, a leading cause of blindness, is often driven by elevated intraocular pressure (IOP), which damages the optic nerve. Transscleral filtration surgery reduces IOP but is frequently complicated by excessive wound healing from Tenon fibroblasts (TFs), impeding aqueous humor absorption. Mitomycin C (MMC), used for over 30 years in ophthalmic surgeries, inhibits TF proliferation but carries significant side effects, including hypotony, blebitis, and endophthalmitis, due to its non-selective cytotoxicity. MMC's inability to entirely prevent fibrosis increases surgical failure risk, often necessitating further interventions like bleb needling. This study investigates whether ascorbic acid (AA) and hydrogen peroxide ($H_2O_2$) can selectively target TFs without damaging scleral fibroblasts (SFs) *in vitro*, using MMC as a benchmark.

### Methods

Primary human TFs and SFs were cultured from patient trabeculectomy tissues. Cells were treated with various concentrations of MMC, AA, or $H_2O_2$. Cytotoxic effects were analyzed via live-cell imaging. Immunocytochemistry and Western Blot assessed catalase expression in both cell types and recombinant catalase was used to validate its protective effect against AA- and $H_2O_2$-induced cell death.

### Results

Short-term exposure (5 min) to 0.02%–0.04% MMC or long-term exposure to 0.00025%–0.001% MMC caused cytotoxicity in TFs and SFs, with SFs dying significantly earlier. In contrast, AA (6–8 mM) selectively induced cell death in TFs without harming SFs. $H_2O_2$ also showed selective cytotoxicity towards TFs. Lower catalase expression in TFs

**Data availability statement:** All relevant data are within the manuscript and its Supporting Information files (PlosoneDataofFigures.xlsx and S1_raw_images.pdf).

**Funding:** This work was partially funded by the "Zentrales Innovationsprogramm Mittelstand" (ZIM) Grant number ZF4603201AW8 of the Federal Ministry for Economic Affairs and Energy, Germany. The funders had no role in study design, data collection and analysis, decision to publish, or preparation of the manuscript.

**Competing interests:** The authors have declared that no competing interests exist.

compared to SFs was determined via Western blot and immunocytochemistry, highlighting a mechanism for this selective effect. Recombinant catalase neutralized the cytotoxic effects of AA and $H_2O_2$ on TFs.

## Conclusions

Unlike MMC, Ascorbic acid and hydrogen peroxide exhibit selective cytotoxicity towards Tenon fibroblasts, which may provide a safer, more targeted approach for preventing fibrosis in glaucoma surgery. Additional *in vivo* studies are needed to explore the clinical applicability of these findings.

## Introduction

In glaucoma implant surgery and classic trabeculectomy, the aqueous humor is directed through a surgically prepared channel into a sub-conjunctival located reservoir called a filtering bleb. The conjunctiva absorbs the aqueous humor, which lowers the elevated intraocular pressure (IOP) to prevent irreparable damage to the optic nerve. However, undesirable excessive wound healing processes, such as fibrosis, occur in numerous cases. Either a fibrotic tenon cyst forms, creating an absorption barrier at the site of the reservoir, or the surgically created fistula is blocked by proliferating cells that produce an extracellular matrix. Both wound-healing processes lead to insufficient absorption of aqueous humor and increased IOP. Wound healing originates from cells of the tenon layer, which spatially separates the sclera from the conjunctiva. To prevent fibrosis of tenon fibroblasts (TFs), parts of the tenon layer are often surgically removed. In addition, mitomycin C (MMC) is applied to inhibit the proliferation of TFs.

Chen and co-workers were the first to propose the intraoperative application of MMC in trabeculectomy with a supposedly poor outcome in an 8-year follow-up study in 1990 [1]. They reported that MMC can prevent excessive wound healing and thus reduce intraocular pressure more efficiently for a more extended period. Jampel confirmed in 1992 that a 5-minute exposure of 0.01% and 0.04% MMC is sufficient to inhibit the proliferation of TFs within 24 h *in vitro* [2]. Intraoperatively, a sponge soaked with 0.02–0.04% MMC is inserted into the prepared conjunctival pocket above the sclera for 2 to 5 min and then rinsed with a saline solution to wash out excess MMC. MMC is a cytostatic drug that penetrates the cells and inhibits DNA replication, thus preventing cell proliferation and inducing apoptosis [3]. Severe DNA damage may lead to DNA mutation and cell death. Therefore, MMC is classified as a carcinogenic substance, and its medical application is controversial. Another disadvantage of MMC is that it does not selectively kill TFs and may induce corneal, scleral, and conjunctival cell apoptosis [4–7].

Severe side effects have been reported, such as postoperative wound healing disorders or the formation of an avascular filtering bleb, hypotony, endophthalmitis, and blebitis [8–11]. Another retrospective study raises doubts that a 2 to 5-minute application of MMC significantly prevents a postoperative tenon cyst formation [12]. It is often reported that an excessive wound-healing process occurs despite the intraoperative application of these cytostatic drugs. A postoperative follow-up treatment, such as a bleb revision, might be necessary. In a bleb needling procedure, MMC or 5-Fluorouracil is often administered subconjunctivally as an adjuvant to restore the function of a fibrosing bleb [9]. The benefit of this postoperative administration of these cytostatic drugs is not undisputed [13]. Despite the above-described controversial effects, MMC has been used in ophthalmic surgery for over 30 years, probably due to a lack of alternatives [1,14–16].

In 1990, Jampel reported that 1.1 mM ascorbic acid (AA), a concentration found in aqueous humor, reduced the plating efficiency of cell suspensions of human Tenon's capsule fibroblasts by an average of 40% within 24 h *in vitro* [17]. Furthermore, he showed that 1.1 mM AA addition reduced cell number by approximately 14% in confluent tenon cultures. He speculated that if AA has a similar effect *in vivo*, it could lead to incomplete wound healing and thus contribute to successful glaucoma filtration surgery.

In contrast to MMC, we aimed to identify compounds that prevent excessive wound healing of TFs without harming scleral fibroblasts (SFs). Therefore, tenon and scleral tissues from six human patients who underwent trabeculectomy were isolated to establish primary cell cultures. These primary cell cultures were further used for drug screening experiments. Here, we investigated the impact of AA and hydrogen peroxide on TFs and SFs *in vitro*.

## Materials and methods

### Cell culture

Tenon and scleral tissues were obtained from six patients who underwent trabeculectomy. The procedures were carried out following the tenets of the Declaration of Helsinki, following the guidelines and regulations of and with the approval of the ethics committee of Hannover Medical School (File number Nr. 8220_BO_K_2018). Patient samples were collected from August 2019 to September 2020. Written informed consent was obtained from all subjects. Some Tenon tissues were excised before the intraoperative application of MMC, while others were removed afterward. Primary TFs were obtained from tenon tissue according to a modified protocol of Przekora et al. [18]. Briefly, tenon tissue was cultured as a tissue attachment explant in Minimum Essential Medium Eagle (MEME, Sigma #M8042, Saint Louis, MI, USA) containing 5% FBS (PAN #P40-39500, Aidenbach, Germany), GlutaMAX™ (Thermo Fisher #35050061, Waltham, MA, USA), Penicillin/Streptomycin (Gibco #3505-038), supplemented with 5 ng ml$^{-1}$ fibroblast growth factor 2 (FGF-2) (Peprotech #AF-100-18B) and 5 μg ml$^{-1}$ insulin (Thermo Fisher #12585014) in a 12-well. An outgrowth of TFs from the tissue could be observed approximately three to four days later. After the initial cell culture, cells were cultivated in the same medium without FGF-2 or insulin. TFs were split at around 90% confluence and were cultured until passage number ten.

Sclera tissue was obtained during the trabeculectomy procedure when a small incision was made in the sclera to alleviate intraocular pressure. SFs obtained from scleral tissue were expanded following the protocol of Seko et al., except that scleral tissue was not cut into small pieces [19]. For SFs, MEME supplemented with 5% FBS, GlutaMAX™, and Penicillin/Streptomycin was used for cultivation. Cell outgrowth was observed after approximately one week. Cell passages were performed at about 90% confluence. TFs and SFs from passages two and eight were used in the following investigations.

### Transforming growth factor-beta 2 treatment

For TGFB2 treatment, $2 \times 10^4$ TFs were seeded in a 24-well plate with 0.5 mL complete medium containing 5% FBS. After 24 h, the medium was exchanged with the medium that contained 2% FBS and 10 ng ml$^{-1}$ human recombinant TGFB2 (Peprotech #100-35B). TFs were cultured for one week, with the medium replaced every four days.

### Immunocytochemistry staining

For ICC staining, $2 \times 10^4$ TFs and SFs were seeded on a 13 mm diameter slide coverslip (Glaswarenfabrik Karl Hecht #41001113, Bavaria, Germany) into each well of a 24-well plate

with 0.5 mL complete medium. Before fixation, cells were washed twice with PBS (Carl Roth #1058.1, Baden-Württemberg, Germany). Cells were immediately fixed with 4% paraformaldehyde (Carl Roth #P087.6) for 20 min at room temperature (RT) and washed twice with PBS for 5 min. Blocking was performed for 1 h at RT with a blocking solution containing 5% goat serum (Millipore #S26-100ml, Burlington, MA, USA), 0.02% Tween®20 (Sigma #P9416) and 0.01% Triton™X-100 (Sigma #X100) in PBS. ACTA2 rabbit mAb (Cell Signaling #19245, Danvers, MA, USA), VIM rabbit mAb (Cell Signaling #5741), FN1 rabbit mAb (Cell Signaling #26836), and Catalase rabbit mAb (Cell Signaling #12980S) were diluted 1:1000 in the blocking solution and cells were incubated overnight at 4°C. Cells were then washed twice with PBS for 10 min at RT and incubated with either 1:500 diluted AlexaFluor™546 goat anti-rabbit (Invitrogen #A11035) or AlexaFluor™488 goat anti-rabbit (Invitrogen #A11034) and 4 µl ml⁻¹ rhodamine-phalloidin (Invitrogen #R415) in PBST (0.1% Tween®20 in PBS) for 2 h at RT. After three washes in PBS for 10 min at RT, the coverslips were mounted upside down on a slide with Roti®-Mount FluorCare DAPI (Carl Roth #HP20.1). Phase-contrast and fluorescence images were recorded with an Observer Z.1 microscope (Carl Zeiss, Baden-Württemberg, Germany) using the ZEN-Blue analysis software (Carl Zeiss).

## Western blot

$10 \times 10^4$ TFs and SFs were cultured in 6-well plates with 2 mL complete medium. Sample preparation, Electrophoresis, blotting, and antibody incubation were performed as previously described by our group [20]. Cells were washed twice with PBS and lysed with 1× Laemmli Sample Buffer (Bio-Rad #1610747, Hercules, CA, USA) containing 1× protease inhibitor cocktail (Cell signaling #5871S). The lysates were mixed with 2-Mercaptoethanol (Sigma-Aldrich #60-24-2), denatured at 70°C for 10 min, centrifuged at 4000 rpm for 2 min, and stored at −20°C. The samples were loaded on Mini-Protean TGX Stain-Free Gels (Bio-Rad #4568094). Briefly, after 80–100 V electrophoresis, the resolved proteins were transferred to an ethanol-activated Mini-size LF PVDF membrane (Bio-Rad #10026934) with Trans-Blot®Turbo™ Transfer System at 1.3 A, 25 V for 7 min. The PVDF membrane was blocked with 5% milk powder (Carl Roth #T145.2) in 1X Tris-buffered saline (TBS) at RT for 1 h, followed by incubation with 1:1000 diluted primary Catalase rabbit mAb (Cell signaling #12980S) overnight at 4°C. After incubation with 1:4000 diluted goat anti-rabbit secondary antibody StarBright™ Blue 700 (Bio-Rad #12004162) at RT for 1 h, the fluorescent signal was detected with Chemi-Doc MP Imaging System (Bio-Rad) at a measurement wavelength of 660–720 nm. ImageLab 6 software (Bio-Rad) was used for normalization and quantification.

## Mitomycin C (MMC) treatment

For the intraoperative application of Mitomycin C (MMC), a concentration of 0.04% MMC was prepared and applied using a sponge. The MMC was left in contact with the tissue for three minutes to allow for adequate absorption. Following the application, the area was washed twice with 10 ml of Hanks' Balanced Salt Solution (HBSS) to remove any residual MMC.

For the co-culture experiments, Tenon fibroblasts (TFs) and scleral fibroblasts (SFs) were seeded in 48-well plates using a 2-well IBIDI-cell culture insert (IBIDI #80209, Bavaria, Germany), with $1 \times 10^4$ cells per well for each cell type. After 24 h, the inserts were removed, and fresh medium containing either 0.02% or 0.04% MMC (Sigma #M4287) in Hank's Balanced Salt Solution (HBSS, Carl Roth #9119.1) was applied to the wells for 5 minutes to mimic intraoperative MMC exposure. Subsequently, the medium was replaced with fresh medium containing 28 nM Hoechst 33342 (Thermo Fisher #H1399) and 2 µg/ml PI (Sigma #P4170) for

live-cell imaging to monitor cytotoxic effects. In addition, for long-term exposure, MMC was directly added to the medium at concentrations of 0.000125%, 0.00025%, or 0.001%, and the cells were exposed and monitored for 90 h to simulate postoperative conditions.

### Ascorbic acid treatment

In monoculture experiments, $3 \times 10^4$ Tenon fibroblasts (TFs) or scleral fibroblasts (SFs) were seeded in 48-well plates with 250 µL of medium containing 2% FBS, 28 nM Hoechst 33342 (Thermo Fisher #H1399), and 2 µg/ml Propidium Iodide (PI) (Sigma-Aldrich #P4170). They were treated with different concentrations of Ascorbic Acid (AA) (Sigma #A0278) ranging from 0 to 8 mM. When 8 mM AA was added, the pH of the medium dropped from 7.7 to 6.8. To rule out the possibility that this pH drop caused cytotoxicity, an HCl control with a pH of 6.8 was included by adding 6.7 µL of 1N HCl (Carl Roth #4625.1) per mL of medium. The medium with HCl was sterilized using a 0.22 µm PES syringe filter (Carl Roth #P668.1). Live-cell imaging was then performed for 24 h.

### Hydrogen peroxide treatment

$1 \times 10^4$ TFs and SFs were seeded and co-cultured as previously. 24 h later, the inserts were taken out, and the medium was changed to fresh medium containing 28 nM Hoechst 33342, 2 µg ml$^{-1}$ PI, different hydrogen peroxide (Sigma #216763-100ML) concentrations (400–600 µM) together with or without 500U ml$^{-1}$ catalase solution (Sigma #SLCJ5891). Live-cell imaging was performed for 24 h.

### Rekombinant catalase treatment

In the co-culture experiments, $1 \times 10^4$ TFs and SFs were seeded as described above. 24 h later, the inserts were taken out, and a fresh medium containing Hoechst, PI, and AA or Hydrogen peroxide was used with or without 400, 500, or 600 U ml$^{-1}$ catalase solution. Live-cell imaging was then performed for 24 h.

### Live-cell imaging

Live cell imaging experiments were performed using the BioTek® Lionheart™ FX Automated Microscope (Agilent Technologies, Santa Clara, CA, USA). The experiment setup and subsequent analysis were performed as previously described by our group using the Gen5 Image Prime 3.05 software [21]. The 48-well culture plate was placed in the humidity chamber with $CO_2$ levels set to 5% and temperature to 37°C. Imaging was carried out in the phase-contrast channel using the 4X PL FL phase objective and laser autofocus. In addition to the phase-contrast images, the DAPI fluorescence filter set was used to detect Hoechst-stained nuclei, and the RFP fluorescence filter was used to detect PI-positive nuclei. Images were recorded for 48 h at 20-min intervals. The acquisition parameters for the phase contrast channel were set to 10 "LED",100 "Integration time" and 4.8 "gain". For DAPI, the values were set to 6 "LED",71 "Integration time" and 24 "gain". For RFP, the values were set to 10 "LED",120 "integration time" and 21 "gain". The relatively high gain settings were used to keep the exposure time as short as possible to minimize photobleaching during live-cell imaging.

### Live-cell imaging analysis

The number of Hoechst-positive objects per image was determined using the cell count option of Gen5 analysis software, as previously described [21]. The analysis tool was briefly used, and the channel "DAPI 377, 4747" was selected under "Primary mask and

Count". The "Threshold" was set to 2000, the background to "Dark", and the options "Split touching objects" and "Fill holes in masks" were selected. In addition, under "Advanced Detection Options", a "Background flattening" with a "Rolling Ball diameter" of 30 μm, an "Image smooth strength" of 2 "Cycles of 3 × 3 average filter", and "Evaluate background on 5% of lowest pixels" were selected. Under "Object selection", a "Min. object size" of 10 μm and a "Max. object size" of 70 μm were specified, and the option "Include primary edge objects" was selected. For the non-co-culture experiments, the option "Analyse entire image" was selected. For the co-culture experiments, the "Plug" tool was used to place a rectangular window on the left side of the image to count the TFs and a rectangular window on the right side to calculate the SFs. Both windows had a size of 750 × 1500 μm. The number of PI-positive objects was calculated using the analysis tool to count dead cells. Therefore, under "Primary mask and Count", the channel "RFP 531, 593" was selected, a threshold of the positive PI signal was set, and the parameters previously described in the DAPI channel were used for further analysis. The data was further analyzed using Microsoft Excel 2010. The number of PI-positive cells was divided by the number of Hoechst-positive nuclei to determine the percentage of dead cells per area for each time point. The mean and standard deviation of biological replicates were calculated from these values with Graph Pad Prism 9.

## Statistical analysis

All experiments were performed in at least three independent replicates, and data were analyzed using Microsoft Excel 2010 and GraphPad Prism 9. A one-way ANOVA was used to determine statistical significance and compare catalase expression between Tenon fibroblasts (TFs) and Scleral fibroblasts (SFs). Post hoc analysis used Tukey's HSD test to compare the groups. Data are presented as mean ± standard deviation (SD) with significance thresholds set at **p < 0.01, and ***p < 0.001.

## Results

### Effects of MMC on TFs *ex vivo/in vitro*

Tenon and scleral explants were collected from six trabeculectomy patients. The explants were transferred into a cell culture medium to allow TF and SF to outgrow, according to a modified protocol of Przekora and Seko [18,19]. Phase contrast images and immunocytochemistry (ICC) staining for vimentin (VIM), actin alpha 2 (ACTA2), and F-actin were obtained from the isolated TFs and SFs (Supplementary Fig S1 in S1 File). Furthermore, it was tested whether TFs transform into myofibroblasts after one week of transforming growth factor beta 2 (TGFB2) exposure, as reported previously by other groups [22–24]. Indeed, TGFB2-treated TFs show a more widespread distribution of VIM within the cytoplasm, an increase of ACTA2, an increase of the extracellular matrix protein fibronectin 1 (FN1), and a resemble of the F-Actin, compared to the untreated TFs (Supplementary Fig S2 in S1 File). Remarkably, two out of six tenon tissue samples were treated intraoperatively with 0.04% MMC for 3 to 5 minutes before the tissue was removed. In contrast, the other four tenon tissues were removed before MMC was applied. However, an outgrowth of proliferating TFs was observed in all tenon tissue samples after 5–7 days, regardless of whether they were treated intraoperatively with MMC (Fig 1a + b, Supplementary Video S1). We successfully expanded and culture TFs up to and beyond passage 10, regardless of intraoperative MMC treatment. This observation suggests that MMC did not exert a lasting antiproliferative effect in ex vivo conditions.

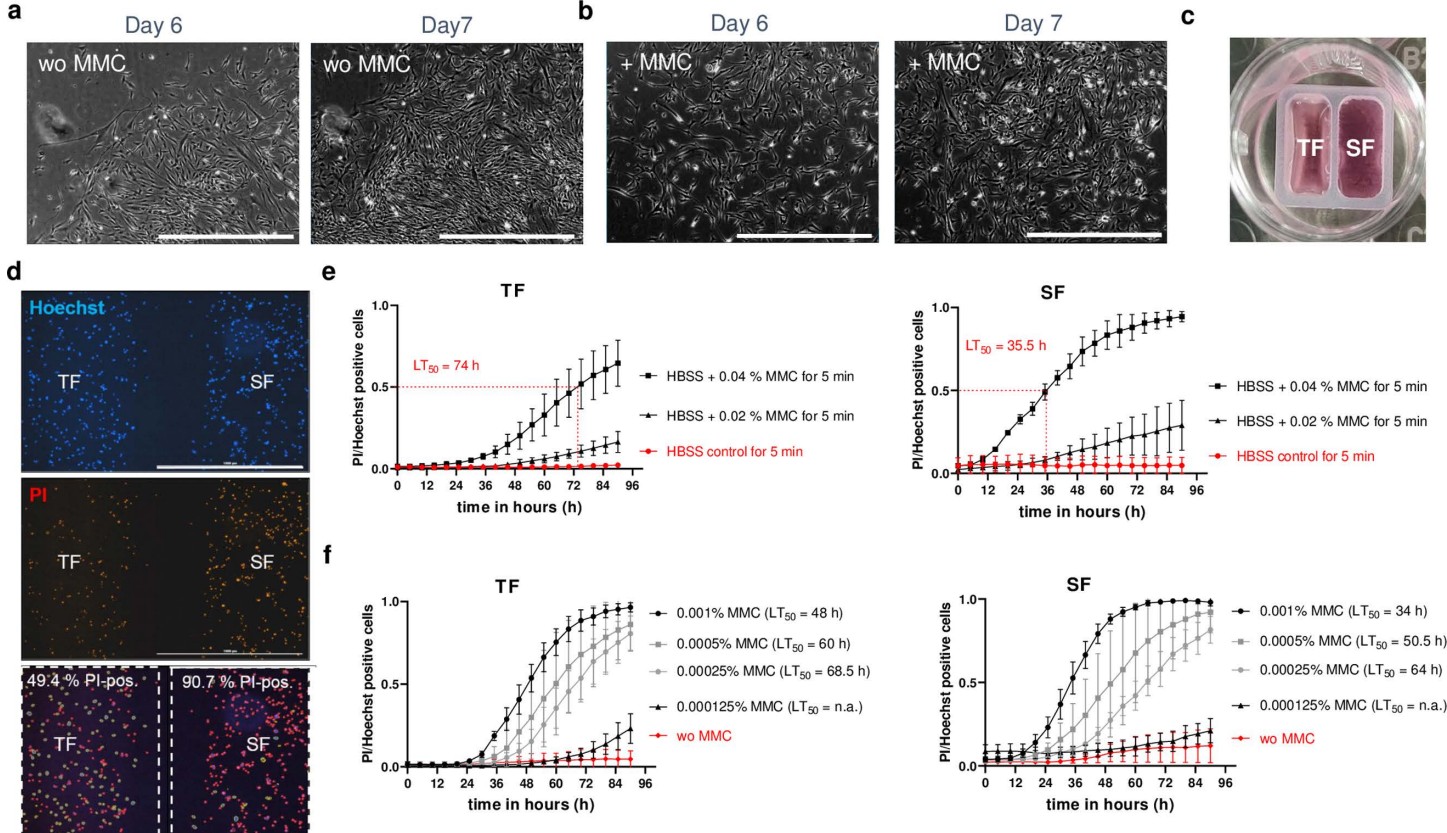

**Fig 1. Cytotoxic effects of MMC on human TFs and SFs *in vitro*.** Representative phase-contrast images showing TF cells at days 6 and 7 after trabeculectomy and initial tenon tissue culture, respectively, treated intraoperatively without (**a**) or with MMC (b). The scale is 1000 µm. (c) Co-culture inserts were used to culture TFs and SFs separately within one well of a 48-well plate. (d) Representative fluorescent pictures of co-cultured TF and SF cells that were treated for 5 min with 0.04% MMC at time point 74 h. During the experiment, cells were stained with Hoechst 33342 (upper row) and PI (middle row) to visualize dead cells. The right and left sections of the image (highlighted by the two white dashed rectangles) were analyzed to quantify the dead TF and SF cells separately. The total number of cells was determined by gating the Hoechst-stained nuclei for the analysis. This mask subsequently identifies PI-positive nuclei by setting a threshold in the red channel. In the lower row, PI-positive nuclei are highlighted in red circles, whereas PI-negative nuclei are highlighted in yellow circles. (e) Cytotoxicity analysis for 90 h of PI- and Hoechst-stained co-cultured TFs (left diagram) and SFs (right diagram) after treatment with 0.02% and 0.04% MMC in HBSS for 5 minutes. HBSS alone for 5 min was used as a control. After MMC treatment and before live-cell imaging, cells were cultured in regular growth media with Hoechst and PI. For the five-minute 0.04% MMC treatment, the LT50, i.e., the time 50% of the cells have died, is also given. (f) Cytotoxicity analysis of PI- and Hoechst-stained co-cultured TFs (left diagram) and SFs (right diagram) that were permanently exposed without or with 0.000125%, 0.00025%, 0.005% or 0.001% MMC for 90 h. Where possible, the $LT_{50}$ for different MMC concentrations was specified here.

## Short-term and long-term exposure to MMC on TFs and SF *in vitro*

Since TFs are in close contact with SFs in the eye, TFs and SFs were cultivated together in a 48-well using a co-culture insert (Fig 1c). To test for a possible cytotoxic effect of MMC in the co-culture cell model, Hoechst 33342 was added to label the cell nuclei, and propidium iodide (PI) to label dead cells. The Hoechst concentration of 28 nM and the acquisition parameters used for live-cell imaging were neither cytotoxic nor phototoxic, as described in previous studies [21,25]. To mimic the intraoperative MMC application *in vitro*, TFs and SFs were treated for 5 minutes without or with 0.02% or 0.04% MMC. After a subsequent medium change with the addition of Hoechst and PI, the number of dead cells was determined over 90 h using a live-cell imager and subsequent image analysis. To determine the percentage of dead cells, the number of PI-positive nuclei was divided by the number of Hoechst-stained nuclei for TFs and SFs separately (Fig 1d), and the calculated lethal time 50 ($LT_{50}$) revealed

that with a five-minute 0.04% MMC application, half of the TFs had died after 74 h. In contrast, the $LT_{50}$ for SFs was only ~ 35.5 h (Fig 1e).

In a further experiment, both cell types were permanently exposed to MMC for 90 h but at lower concentrations between 0.001% and 0.000125% to simulate potential effects in cases where MMC is insufficiently washed out. The $LT_{50}$ values for each MMC concentration confirmed the tendency for SFs to die faster from MMC than TFs (Fig 1f). Representative time-lapse videos of the co-cultures exposed to either 0.04% MMC for 5 min or 0.001% MMC permanently are shown in the Supplementary Video S2.

## Impact of AA on TFs and SFs *in vitro*

Since Jampel demonstrated in 1990 that 1 mM AA inhibits the viability of TFs in vitro, we wanted to test the effect of AA on both cell types. For this approach, confluent TFs or SFs were exposed to different AA concentrations between 0 and 8 mM for 24 h. Notably, the pH value of the 8 mM AA treatment condition decreased from 7.7 to 6.8. To exclude effects caused by pH decrease, an additional control was included with a medium adjusted to pH 6.8 using 1N hydrochloric acid (HCl). The percentage of dead TFs (left panel) and SFs (right panel) within 24 h was determined by dividing the number of PI-positive cells by the total number of Hoechst-positive cells (Fig 2A). Data reveals that AA concentrations of 6 and 8 mM resulted in complete cell death of TFs within 24 h. No significant cell death was observed in the untreated cells or HCl control. Furthermore, no significant cytotoxic effect of AA on SFs was observed (Supplementary Video S3). No AA-induced cytotoxicity was observed in sclera cells after the 24 h window. Therefore, subsequent LCI analyses were performed in the 24-h time window.

To evaluate the possibility that AA-induced cell death was caused by nuclear staining with Hoechst, we treated TFs with 0 and 6 mM AA in combination with and without Hoechst in the medium. Live-cell imaging analysis showed that Hoechst alone did not cause cytotoxicity in the TFs within 24 h, while the TFs treated with 6 mM AA died after about 12 h without any synergistic effect caused by Hoechst staining (Supplementary Figure S3 in S1 File). To further exclude whether AA-induced cell death of TFs was possibly patient-specific or coincidental, we tested AA on TFs and SFs of six trabeculectomy patient samples. Therefore, we treated TFs and SFs from six trabeculectomy patients in monoculture experiments without or with 6 mM and 8 mM AA. In all 6 patient samples, AA concentrations between 6 and 8 mM were cytotoxic for TFs but harmless for SFs (Fig 2b, c).

Since the tenon cell layer is located directly above the sclera *in vivo*, we wanted to investigate whether signaling molecules or free radicals might be released by AA-induced death of TFs, which could impair the viability of the SFs. For this purpose, we co-cultured TFs and SFs in one well of a two-chamber cell culture set and incubated them for 24 h. The insert was removed, spatially separating both cell types, and treated with or without 6 mM AA with PI and Hoechst. In co-cultures, we observed that TFs ultimately died within 24 h in the presence of 6 mM AA, whereas no significant cytotoxicity was observed for the SFs (Fig 2d, e, Supplementary Video S4).

## Catalase expression in TFs and SFs

Next, we wanted to investigate how AA selectively attacks the TFs without damaging the SFs. Numerous studies have shown that AA sometimes initiates the $Fe^{3+} \lozenge Fe^{2+}$ redox cycle in the Fenton reaction. $Fe^{2+}$ further reacts with hydrogen peroxide, increasing toxic hydroxyl radicals that can irreparably damage the cells [26,27]. Consequently, the concentration of hydroxyl radicals formed depends on the amount of free $Fe^{2+}$ and the concentration of hydrogen peroxide. The hydrogen peroxide concentration is regulated by endogenous catalase expression.

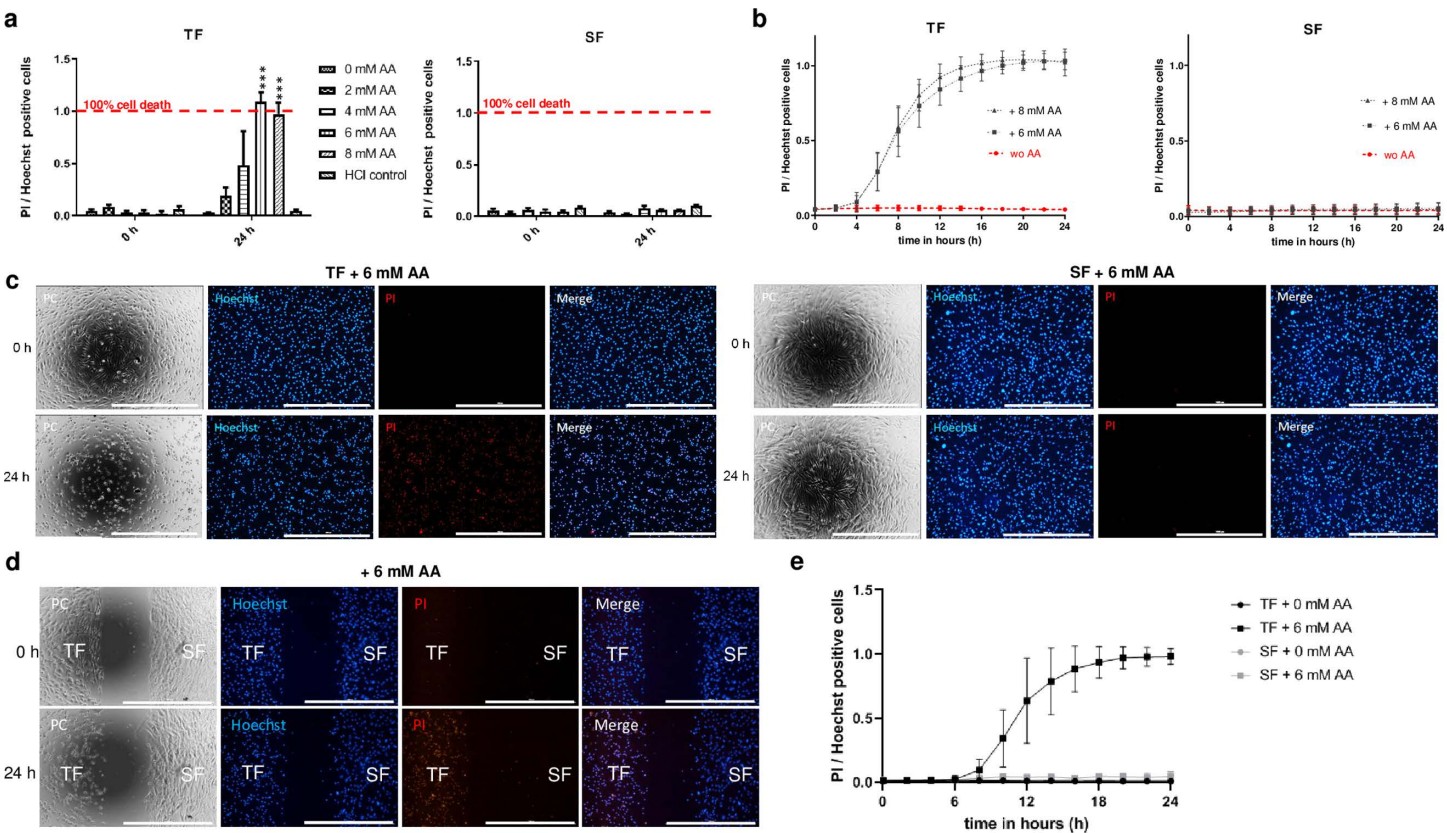

**Fig 2. Low concentrations of AA selectively kill human TFs without harming SFs *in vitro*.** (a) The two bar diagrams show the percentage of dead TFs (left side) and SFs (right side), which were exposed to 0, 2, 4, 6, and 8 mM AA or HCl control at time point zero and after 24 h. (b) Cell toxicity analysis of monocultured TFs (right diagram) and SFs (left diagram) that were treated without or with 6 mM or 8 mM AA for 24 h. For data collection, live cell analysis was performed three times on TFs and SFs obtained from six trabeculectomy patients. (c) Representative phase-contrast and Hoechst and PI fluorescence images of monocultured TFs (left panel) and SFs (right panel) with 6 mM AA for 0 h (upper row) and 24 h (lower row). The scale bar represents 1000 μm. (d) Representative phase-contrast and Hoechst and PI fluorescence images of co-cultured TFs and SFs treated with 6 mM AA for 0 h (upper row) and 24 h (lower row). The scale bar represents 1000 μm. (e) Cell toxicity analysis of co-cultured TFs and SFs treated without or with 6 mM AA for 24 h.

Immunofluorescence staining revealed that both cell types express catalase within the cytoplasm (Fig 3a). However, a Western blot analysis of TFs and SFs from three patients showed that the catalase protein levels are significantly reduced in TFs compared to SFs (Fig 3b, c).

## Recombinant catalase abolishes AA- and hydrogen peroxide-induced cell death

A further co-culture experiment was performed to verify whether the AA-induced cell death of the TFs was due to their low catalase activity. Therefore, TFs and SFs were treated with 4 mM AA with and without 500 Units of recombinant catalase. Indeed, the addition of catalase reversed AA-induced cell death of TFs (Fig 4a + b, Supplementary Video S5). Finally, it was tested whether the reduced catalase activity of TFs leads to TFs reacting more sensitively to hydrogen peroxide than their neighboring SFs. Therefore, a co-culture experiment was performed on TFs and SFs with and without recombinant catalase in combination with hydrogen peroxide concentrations between 400 and 600 μM (Fig 4c + d, Supplementary Fig S4 in S1 File, Supplementary Video S6). Here, the SFs were more robust to hydrogen peroxide exposure than TFs. Again, this effect could be reversed by the addition of recombinant catalase.

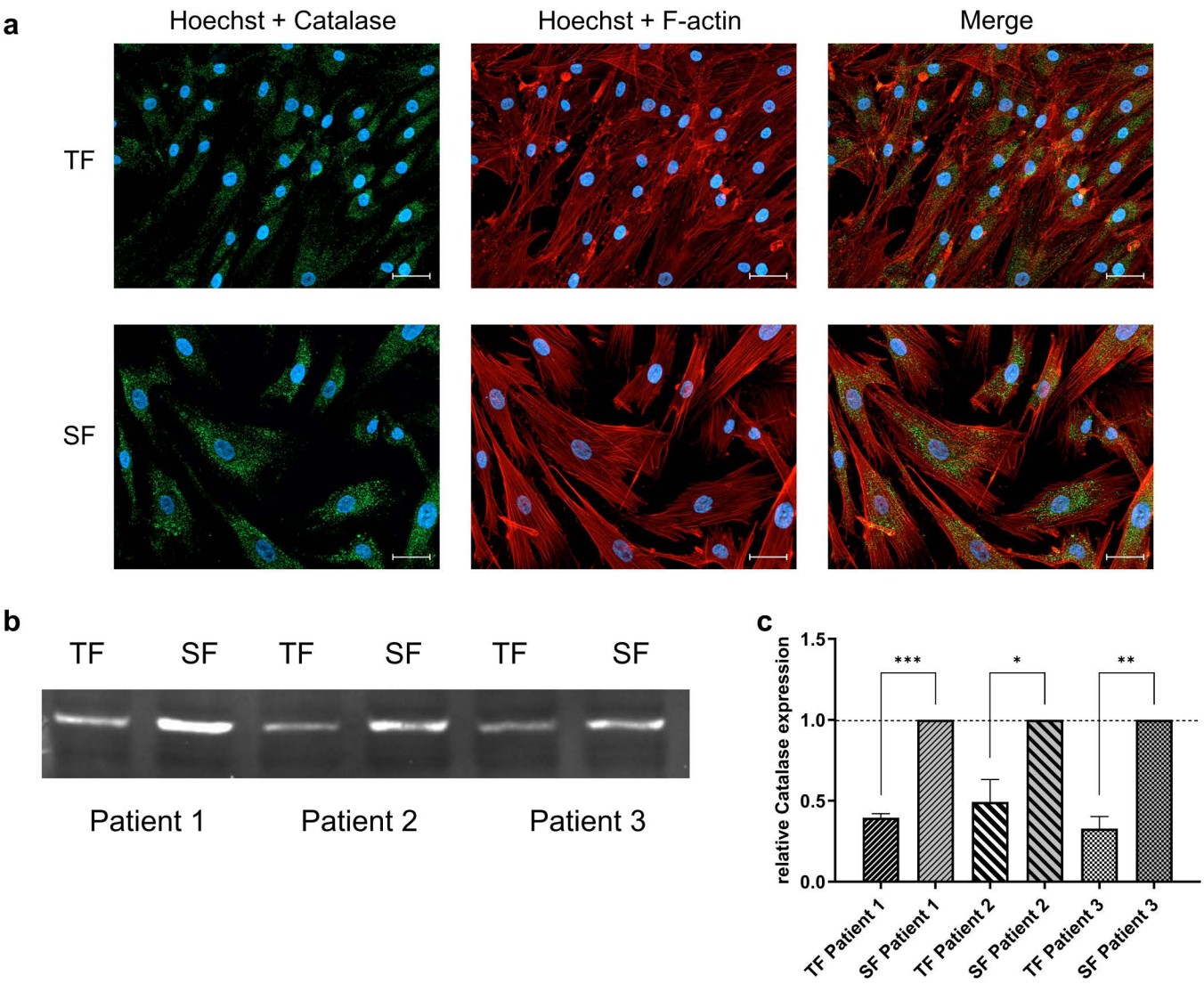

**Fig 3. Unveiling a potential weak spot: TFs have lower catalase expression than SFs.** (a) Representative ICC-staining showing catalase expression (green), F-Actin (red), and DAPI (blue) of TFs (upper row) and SFs (lower row). (b) Representative fluorescence Western blot showing different catalase protein expressions in TFs and SFs obtained from three technical and three biological replicates. (c) Normalized, quantitative catalase protein expression. The densitometric ratio values were previously normalized to total protein from three technical replicates/ passage numbers (original data shown in Supplementary Figure S5 in S1 File) and further normalized to the SF samples. An unpaired two-tailed t-test evaluated differences.

## Discussion

In this *in vitro* study, we partially confirmed the antiproliferative effect of MMC as described by Jampel [2]. However, our live-cell analyses revealed that this effect is driven by MMC's cytotoxicity rather than a specific inhibition of Tenon fibroblast (TF) proliferation. Jampel's original study explored a 24-h time window, but we found that cytotoxicity from the 5-minute 0.04% MMC exposure only became evident after 36 h, with the $LT_{50}$ reached at 72 h (Fig 1e, Supplementary Video S2). This delayed cytotoxicity raises concerns, as we also observed that MMC preferentially kills scleral fibroblasts (SFs) faster than TFs (Fig 1e + f). This non-selective action of MMC is consistent with previous reports, which have documented its

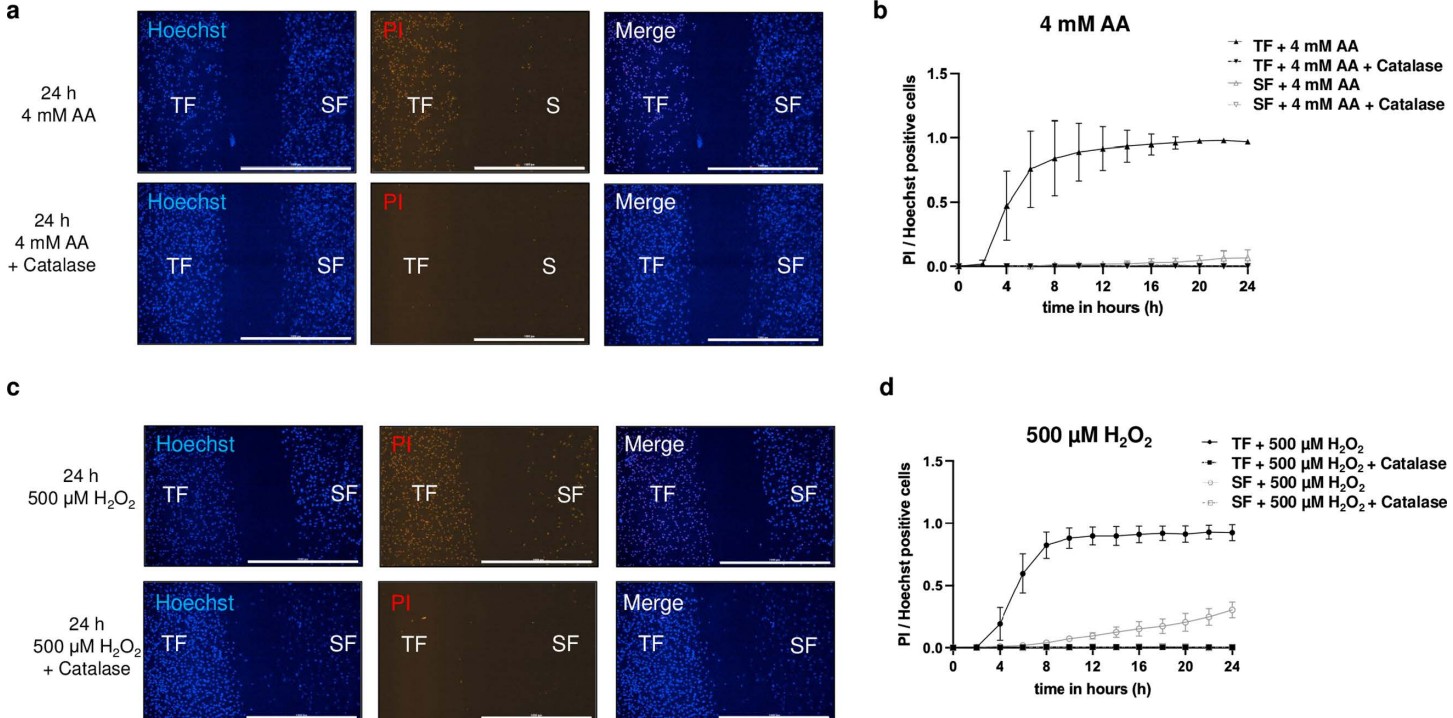

**Fig 4. Catalase supplementation prevents AA-induced and hydrogen peroxide-induced cell death of TFs *in vitro*.** (a) Co-culture of TFs (left) and SFs (right) that were exposed to 4 mM AA without (upper row) or with catalase (lower row) for 24 h. The nuclei were stained with Hoechst, and PI was added to the medium to visualize dead cells. The scale represents 1000 μm. (b) The number of dead cells was calculated from the number of PI-positive cells against the total number of Hoechst-positive cells treated without or with 4 mM AA without or with catalase for 24 h in 2-h intervals. The data shows the mean number of dead cells from TFs and SFs obtained from three patients. (c) Co-culture of TFs (left) and SFs (right) that were exposed to 500 μM hydrogen peroxide without (upper row) or with catalase (lower row) for 24 h. The nuclei are stained with Hoechst, and dead cells are stained with PI. The scale represents 1000 μm. (d) The relative number of dead TFs and SFs was calculated in 2-h intervals from the number of PI-positive cells against the total number of Hoechst-positive cells treated without or with 500 μM hydrogen peroxide without or with catalase for 24 h.

effects on scleral, corneal, and conjunctival cells [4,7], as well as complications such as cystic bleb formation, hypotony, or endophthalmitis findings [4,8,9], we aimed to identify compounds that selectively induce cytotoxicity in TFs without harming SFs, thus potentially reducing fibrosis-related complications post-glaucoma surgery.

The rationale for investigating AA and hydrogen peroxide ($H_2O_2$) was driven by the need for more selective alternatives to MMC to spare adjacent tissues while targeting TFs to mitigate the excessive wound healing that complicates glaucoma surgery.

AA was initially identified during a compound screening in which it unexpectedly accelerated TF death, prompting us to explore its potential further. We hypothesized that AA could induce selective cytotoxicity via oxidative stress, primarily through the Fenton reaction, wherein AA, acting as a pro-oxidant, generates reactive oxygen species (ROS). Our results support this hypothesis, showing that 6–8 mM AA concentrations selectively killed TFs without affecting SFs (Fig 2a–c, Supplementary Video S3). Interestingly, Jampel's 1990 study also identified AA affecting TFs survival at 1.1 mM [17]. His method involved treating TF cultures derived from human tenon cysts with AA, detaching the cells using trypsin, and re-plating them to quantify the number of reattaching cells. However, we believe this approach, which assesses cell death based on non-attached cells, may not fully capture the subtler cytotoxic effects, as detaching cells with trypsin could interfere with viability measurement. Therefore, our study extends these findings by demonstrating that higher concentrations induce apoptosis rather than merely inhibit proliferation.

The co-culture experiments further demonstrated the selective cytotoxicity of AA on TFs. TFs in monoculture began dying after 5 h and were entirely dead by 12 h at 6 mM AA. In co-culture with SFs, TF death was delayed, suggesting that SFs may mitigate but cannot entirely prevent TF death under these conditions (Fig 2d + e, Supplementary Video S4).

Typically, AA acts as an antioxidant and can help neutralize reactive oxygen. However, in certain situations, AA can support the production of hydroxyl radicals by reducing $Fe^{3+}$ to $Fe^{2+}$. $Fe^{2+}$, in combination with hydrogen peroxide, can initiate the Fenton reaction in which hydroxyl radicals are formed, resulting in irreparable damage to the cells. The cell death caused by the Fenton reaction is also known as ferroptosis and is being investigated as a possible therapeutic approach for killing tumor cells [28,29]. According to the manufacturer, no iron ions exist in the MEM Medium we used for the cell culture experiments. However, fetal bovine serum (FBS) has been reported to contain ferritin, a protein complex that binds iron ions [30].

The selective effect of AA on TFs may be related to the fact that TFs have a higher hydrogen peroxide level than SFs. The hydrogen peroxide level of the cells is regulated by the enzyme catalase, which neutralizes hydrogen peroxide by cleaving 2x $H_2O_2$ into 2x $H_2O$ and $O_2$. Consequently, low catalase expression could increase the cell's hydrogen peroxide concentration, resulting in more hydrogen peroxide forming lethal hydroxyl radicals during the AA-induced Fenton reaction. Therefore, we compared the catalase expression of TFs and SFs from three patients using ICC staining and Western blot quantification. Western blot analysis revealed that catalase expression was reduced by approximately 50% in TFs compared to SFs (Fig 3b + c). The selective cytotoxicity of AA on TFs could be entirely abolished by adding recombinant catalase (Fig 4a + b, Supplementary Video S5). In addition, we show that TFs were more sensitive to hydrogen peroxide in the medium than SFs, presumably due to their reduced catalase expression. Here, too, recombinant catalase could prevent hydrogen peroxide-induced cell death (Fig 4c + d, Supplementary Video S6). Thus, the reduced catalase expression of TFs represents a weak point that potentially opens up new therapeutic options for IOP-lowering glaucoma surgery.

One limitation of the study is that, for ethical reasons, we could not obtain corneal and conjunctival tissue from the trabeculectomy patients, as these were not removed during surgery. Although we were only able to test the influence of AA on TFs and SFs, the question arises as to what extent AA influences the viability of corneal and conjunctival cells. Treatment with AA or its derivatives has been extensively studied in rabbit eyes, but to our knowledge, there is still too little data for humans. For corneal injuries of the rabbit eye, a ten percent AA solution applied as eye drops showed significantly improved wound healing after one week [31]. Additionally, Chen et al. reported that AA promotes corneal epithelial wound healing in mice by stimulating the stemness of corneal progenitor cells [32]. Other groups showed that AA eye drop solution could inhibit induced corneal neovascularization in rats and rabbits [33,34]. The reports mentioned above used a much higher concentration of AA than we used in our study, indicating that the AA concentration we used (8 mM corresponds to a 0.15% solution) is unlikely to affect the viability of corneal or conjunctival cells. Another limitation of this study is that we could not collect clinical data for ethical reasons, as mitomycin C is considered the gold standard for IOP-reducing surgeries in Germany.

In Summary, our data indicate that AA and hydrogen peroxide can prevent excessive wound healing of TFs and, unlike MMC, do not damage SFs significantly *in vitro*. The extent to which both can be applied intraoperatively or postoperatively *in vivo*, for example, during bleb revision, and whether additional iron ions are required *in vivo* to initiate the Fenton reaction by AA will require further investigations.

## Supporting information

**S1 File: S1 Figure. Phase-contrast images and ICC-Staining for ACTA2, VIM, FN1, and F-Actin on human TFs and SFs.** TFs (upper row) and SFs (lower row) at passage # 3 were cultured for 1 week and ICC-stained for ACTA2, VIM, or Phalloidin-Red to visualize F-Actin. Nuclei were stained with DAPI, and the scale bar represents 50 μm. **S2 Figure. ICC-staining of human TFs and their ability to induce TGFB2-mediated endothelial-to-mesenchymal transition (EndoTM).** Representative ICC results from human TFs treated without (upper row) or with recombinant TGFB2 (lower row) for one week. Cells were immunostained for ACTA2, FN1, VIM, or Phalloidin-Red to visualize F-Actin. The scale bar represents 50 μm. **S3 Figure. Hoechst 33342 does not impact the AA-induced cell death.** Cytotoxicity analysis of TFs (left) and SFs (right) exposed to 6 mM AA in combination with or without 28 nM Hoechst 33342. The y-axis shows the total number of PI-positive nuclei. **S4 Figure. Catalase prevents hydrogen peroxide-induced cell death in both SFs and TFs.** The relative numbers of dead TFs and SFs were calculated in two-hour intervals by dividing the number of PI-positive cells by the number of Hoechst-positive cells treated without or with 400 μM (left) or 600 μM (right) hydrogen peroxide without or with 500 Units Catalase for 24 h. **S5 Figure: A-C) Raw WB images used for Fig. 3 b and WB quantification**.
(PDF)

**S2 File. Plos one Data of Figures**
(XLSX)

**Supplementary Video S1. Phase contrast time-lapse image videos of TFs between days 6 and 7 after initial culture were treated intraoperatively without or with 0.02% MMC for 3 minutes. The scale bar presents 1000 μm.**
(MP4)

**Supplementary Video S2. Time-lapse image videos showing co-cultured TFs and SFs treated *in vitro* either with 0.04% MMC for 5 minutes (left video) or permanently exposed to 0.001% MMC for 90 hours (right video). Hoechst 33342 and PI were added to visualize the nuclei and numbers of dead cells. PI-negative nuclei are masked with yellow circles, whereas PI-positive nuclei are highlighted in red circles. The scale bar presents 1000 μm.**
(MP4)

**Supplementary Video S3. Time-lapse video showing mono-cultured TFs and SFs treated without or with 6 mM AA for 36 h. In addition to the phase-contrast recording, the Hoechst-stained nuclei are shown. The scale bar presents 1000 μm.**
(MP4)

**Supplementary Video S4. Time-lapse videos showing co-cultured TFs and SFs exposed to 6 mM AA for 24 h. In addition to the phase-contrast recording, fluorescence recordings of the Hoechst-stained nuclei, the PI-stained nuclei, and the merged Hoechst and PI recordings are shown. The scale bar presents 1000 μm.**
(MP4)

**Supplementary Video S5. Time-lapse videos showing co-cultured TFs and SFs exposed to 4 mM AA (left video) or combined with 500 U recombinant catalase (right video) for 24 h. Hoechst 33342 and PI were added to visualize the nuclei and numbers of dead cells. PI-negative nuclei are masked with yellow circles, whereas PI-positive nuclei are highlighted in red circles. The scale bar presents 1000 μm.**
(MP4)

**Supplementary Video S6.** **Time-lapse videos showing co-cultured TFs and SFs exposed to 500 μM hydrogen peroxide (left video) or combined with 500 U recombinant catalase (right video) for 24 h. Hoechst 33342 and PI were added to visualize the nuclei and numbers of dead cells. PI-negative nuclei are masked with yellow circles, whereas PI-positive nuclei are highlighted in red circles. The scale bar presents 1000 μm.**
(MP4)

## Author contributions

**Conceptualization:** Heiko Fuchs, Xiaonan Hu.

**Data curation:** Heiko Fuchs, Xiaonan Hu, Roland Meister, Yuqing Huang, Martin C. Bartram, Amelie Pielen, Bernd Junker, Jan Tode.

**Formal analysis:** Heiko Fuchs, Xiaonan Hu, Roland Meister.

**Funding acquisition:** Carsten Framme.

**Investigation:** Heiko Fuchs.

**Methodology:** Heiko Fuchs.

**Resources:** Martin C. Bartram, Amelie Pielen, Bernd Junker, Jan Tode.

**Supervision:** Heiko Fuchs, Carsten Framme.

**Validation:** Xiaonan Hu.

**Writing – original draft:** Heiko Fuchs, Xiaonan Hu.

**Writing – review & editing:** Heiko Fuchs, Xiaonan Hu, Roland Meister, Yuqing Huang, Martin C. Bartram, Amelie Pielen, Bernd Junker, Jan Tode, Carsten Framme.

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
