## [Decision Letter · Decision Letter 0]

9 Sep 2024

PONE-D-24-26851

Selective Cytotoxicity of Ascorbic Acid and Hydrogen Peroxide on Primary Human Tenon Fibroblasts without harming Scleral fibroblasts in vitro

PLOS ONE

Dear Dr. Fuchs,

Thank you for submitting your manuscript to PLOS ONE. After careful consideration, we feel that it has merit but does not fully meet PLOS ONE’s publication criteria as it currently stands. Therefore, we invite you to submit a revised version of the manuscript that addresses the points raised during the review process.

We look forward to receiving your revised manuscript.

Kind regards,

Dr. Sanaz Alaee

Academic Editor

PLOS ONE

Journal Requirements:

   "This work was partially funded by the "Zentrales Innovationsprogramm Mittelstand" (ZIM) Grant number ZF4603201AW8 of the Federal Ministry for Economic Affairs and Energy, Germany."

Additional Editor Comments:

Expert reviewers in the field have evaluated your submission and found it not up to PLOS ONE publication standards. Please see their detailed comments.

Thank you for submitting your manuscript to PLOS ONE. After a thorough review, we have determined that a major revision is required for your manuscript to meet our publication criteria. Therefore, we request that you make significant revisions to your manuscript to be reconsidered for publication.

Reviewers' comments:

Reviewer's Responses to Questions

**Comments to the Author**

1. Is the manuscript technically sound, and do the data support the conclusions?

Reviewer #1: Yes

Reviewer #2: Yes

2. Has the statistical analysis been performed appropriately and rigorously? 

Reviewer #1: Yes

Reviewer #2: Yes

3. Have the authors made all data underlying the findings in their manuscript fully available?

Reviewer #1: Yes

Reviewer #2: Yes

4. Is the manuscript presented in an intelligible fashion and written in standard English?

Reviewer #1: Yes

Reviewer #2: Yes

5. Review Comments to the Author

Reviewer #1: The manuscript presents a valuable study on the selective cytotoxicity of Tenon fibroblasts (TFs) using ascorbic acid (AA) and hydrogen peroxide (H₂O₂). While the research is commendable, there are several areas that need improvement for better clarity and structure.

Methods and Results Sections: There is a significant overlap between the methods and results, particularly in the co-culture experiment section. Separating these sections will improve readability. I suggest clearly defining the methodology first and then explaining the results obtained.

Rationale for Compound Selection: The choice of using hydrogen peroxide and ascorbic acid requires more explanation. Why were these compounds chosen, and what specific advantage do they offer over Mitomycin C (MMC)? The discussion should delve into the potential roles of these compounds in oxidative stress or ferroptosis.

Objective of MMC Discussion: The manuscript includes MMC in the introduction, but its connection to the study is unclear. Is the goal to find an alternative to MMC, or is it a comparative analysis? The objective should be clearly defined.

High Passage Number for TFs: Using passage 10 Tenon fibroblasts raises concerns since higher passages might behave differently than early passage cells. A justification for choosing this passage number is required to address potential concerns about altered cell behavior.

Passage Number for SFs: The passage number of scleral fibroblasts (SFs) is not provided. To ensure reproducibility, this should be mentioned and explained, particularly to ensure the use of comparable passages.

TGF-β Inclusion: The rationale for using Transforming Growth Factor-beta (TGF-β) is unclear. The authors should explain the reason for incorporating TGF-β, especially given its role in fibrosis.

Omission in Methods: Line 241 references treatments with MMC, but this is missing from the methods section. All treatments and procedures should be clearly outlined for better transparency.

Statistical Methods: The use of a t-test in comparing more than two groups is not appropriate. A better statistical approach, such as ANOVA, should be applied when comparing three groups.

pH Measurement: Only the pH of the 8 mM AA solution is reported. It is important to confirm whether other concentrations maintained a stable pH, as this could affect the results.

Summary Revision: The summary contains information that doesn't directly reflect the study's results. The authors should revise the summary to focus only on their experimental findings.

Reviewer #2: This study offers valuable insights into the potential of using ascorbic acid (AA) and hydrogen peroxide (H₂O₂) to prevent fibrosis following glaucoma filtration surgery. However, several aspects of the manuscript need revision to improve clarity and focus.

Clear Separation of Methods and Results: There is a mix of methods and results, particularly in the co-culture experiment section involving Tenon fibroblasts (TFs) and scleral fibroblasts (SFs). These sections should be kept distinct to enhance the structure and flow of the manuscript.

Explanation for the Use of AA and H₂O₂: The rationale for choosing ascorbic acid and hydrogen peroxide is insufficiently explained. The authors should elaborate on why these compounds were selected and their advantages over MMC, particularly their role in inducing oxidative stress or ferroptosis.

MMC Discussion Clarification: It is unclear if the goal of the study is to replace MMC or simply to compare it with AA and H₂O₂. The objective must be stated clearly in the introduction to avoid confusion.

Passage Number for TFs and SFs: The use of passage 10 TFs is concerning, as high-passage fibroblasts may differ from early-passage cells in terms of behavior. Additionally, the passage number for scleral fibroblasts (SFs) is missing and should be included in the methods for consistency and reproducibility.

Statistical Analysis Issues: In some comparisons where more than two groups are analyzed, a t-test is used, which is inappropriate. A more suitable statistical method like ANOVA should be used.

Discrepancy in AA Results: The authors should address the discrepancy between their results and Jampel et al.'s study regarding the cytotoxicity of AA. A discussion on why the current study required higher concentrations of AA would help clarify the differences between findings.

6. PLOS authors have the option to publish the peer review history of their article (what does this mean? ). If published, this will include your full peer review and any attached files.

**Do you want your identity to be public for this peer review?** For information about this choice, including consent withdrawal, please see our Privacy Policy .

Reviewer #1: No

Reviewer #2: No

---

## [Author Response · Author response to Decision Letter 1]

31 Oct 2024

Dear Reviewers,

We appreciate the thorough evaluation of our manuscript and the constructive feedback provided. Your comments have highlighted key areas for improvement. We have combined responses to overlapping comments to avoid redundancy and address each point comprehensively.

Response to both reviewers:

Reviewer #1: Methods and Results Sections: There is a significant overlap between the methods and results, particularly in the co-culture experiment section. Separating these sections will improve readability. I suggest clearly defining the methodology first and then explaining the results obtained.

Reviewer #2: Clear Separation of Methods and Results: There is a mix of methods and results, particularly in the co-culture experiment section involving Tenon fibroblasts (TFs) and scleral fibroblasts (SFs). These sections should be kept distinct to enhance the structure and flow of the manuscript.

Response:

We thank the reviewers for this comment. We acknowledge the suggestion to improve the separation of methods and results. However, in the case of the co-culture experiment, we intentionally provided detailed descriptions of the experimental setup in the results section to ensure that readers unfamiliar with IBIDI co-culture inserts can easily understand how both Tenon and scleral cells were spatially separated within one well. This co-culture system mimics the natural environment of these cells in the eye, where Tenon and scleral fibroblasts are neighboring but not mixed. Since MMC exposure occurs for both cell types simultaneously during surgery, we felt it was crucial to present this information alongside the results to avoid confusion and unnecessary back-and-forth between the methods and results sections.

Furthermore, we included images of the IBIDI inserts and the methodology for live-cell imaging to demonstrate how both cell types were analyzed separately. This made the data more comprehensible and underscored our commitment to transparency. We believe that this approach offers a more transparent narrative for the reader, especially those less familiar with such experimental setups, and reassures them of the validity of our findings.

Reviewer #1: Rationale for Compound Selection: The choice of using hydrogen peroxide and ascorbic acid requires more explanation. Why were these compounds chosen, and what specific advantage do they offer over Mitomycin C (MMC)? The discussion should delve into the potential roles of these compounds in oxidative stress or ferroptosis.

Reviewer #2: Explanation for the Use of AA and H2O2: The rationale for choosing ascorbic acid and hydrogen peroxide is insufficiently explained. The authors should elaborate on why these compounds were selected and their advantages over MMC, particularly their role in inducing oxidative stress or ferroptosis.

Response:

We appreciate the opportunity to explain further the reasoning behind our selection of Ascorbic Acid (AA) and Hydrogen Peroxide (H2O2). During a compound screening as part of the Glaukom project, we aimed to identify substances that selectively induce cell death in Tenon fibroblasts (TFs) without affecting neighboring Scleral fibroblasts (SFs). The goal was to find an alternative to Mitomycin C (MMC), which does not selectively target TFs and can lead to postoperative complications.

During this screening, we identified inhibitors that triggered autophagy selectively in TFs without harming SFs. Autophagy, a process of controlled cell degradation, appeared to be a promising target. We conducted an in-silico analysis, which revealed that Ascorbic Acid had been reported to inhibit autophagy in specific cell types. Consequently, we hypothesized that AA could counteract the autophagic death induced by these inhibitors. Surprisingly, our experiments showed the opposite: AA exhibited a synergistic effect, accelerating the death of TFs instead of rescuing them. This unexpected outcome led us to investigate AA further as a stand-alone agent. Since the findings from the autophagy screening are still unpublished and will soon be submitted for publication, we cannot disclose the specific inhibitors identified. We kindly ask for your understanding on this point.

As for Hydrogen Peroxide's selection, it resulted from our observation that TFs express lower catalase levels than SFs, making them more vulnerable to oxidative stress. Our Western blot analysis confirmed this differential expression, prompting us to test whether H2O2 could induce selective cytotoxicity by exploiting the reduced catalase expression in TFs. Indeed, we found that H2O2 selectively damaged TFs while sparing SFs.

As a result, the reduced catalase expression in TFs represents an Achilles' heel, opening the door for novel therapeutic approaches to glaucoma surgery. By contrast, MMC lacks this selectivity and is associated with severe complications, such as hypotony, blebitis, and scleral thinning. We are confident that AA and H2O2 offer significant advantages in terms of safety and selectivity and could be promising alternatives for preventing intraoperative or postoperative fibrosis.

Reviewer #1: Objective of MMC Discussion: The manuscript includes MMC in the introduction, but its connection to the study is unclear. Is the goal to find an alternative to MMC, or is it a comparative analysis? The objective should be clearly defined.

Reviewer #2: MMC Discussion Clarification: It is unclear if the goal of the study is to replace MMC or simply to compare it with AA and H2O2. The objective must be stated clearly in the introduction to avoid confusion.

Response:

We appreciate the reviewers' comments and the need for clarification regarding the role of MMC in this study. To address the concern, we would like to clearly state that the primary objective of our study is to explore whether Ascorbic Acid (AA) and Hydrogen Peroxide (H2O2) can offer a more selective alternative to MMC in targeting Tenon fibroblasts, without causing damage to surrounding cell types such as scleral fibroblasts. While MMC is included for comparison, the focus of our investigation is on identifying safer and more targeted treatments for glaucoma surgery.

In Germany, MMC is widely regarded as the gold standard for preventing fibrosis after glaucoma filtration surgery, despite its well-documented side effects, such as hypotony, blebitis, and endophthalmitis. MMC's non-selective mechanism damages Tenon fibroblasts and surrounding cells, including scleral fibroblasts. This lack of specificity contributes to complications that can significantly impact patient outcomes. Given these risks, there is an urgent need for safer alternatives.

One notable practical advantage of AA is its affordability, but this could also present a challenge in securing funding for clinical off-label studies. Due to the low financial return on investment, sponsors may have little incentive to fund trials, which could delay AA’s adoption in clinical practice. Despite these economic challenges, we believe AA’s potential benefits, including its safety profile and effectiveness, warrant further investigation.

To simulate the intraoperative application of MMC as commonly used in glaucoma surgeries, we employed a 5-minute exposure in our experiments. This exposure duration aligns with clinical practice, where MMC is typically applied at concentrations of 0.02%-0.04% for 2-5 minutes. By using this approach in our co-culture experiments with both Tenon and Scleral fibroblasts, we closely mimicked surgical conditions to examine the differential effects of MMC on these two cell types.

Our findings, including the Lethal Time 50 (LT50) analysis, demonstrated a significant difference in cell death kinetics between scleral fibroblasts (SFs) and Tenon fibroblasts (TFs). Specifically, scleral cells died faster than Tenon cells when exposed to MMC, reinforcing that MMC, while effective, lacks the cell selectivity required for targeted therapy in glaucoma surgery.

By employing a co-culture setup, we were able to precisely monitor the effects of MMC on both cell types under identical conditions. This experiment provided valuable insights into MMC's non-selective cytotoxicity and underscored the need to explore alternative agents, such as AA and H2O2, that could selectively target Tenon fibroblasts without harming surrounding tissues. These findings will be clearly articulated in the revised manuscript.

To avoid any misunderstanding and to clarify the study's objective, we have changed the title of our manuscript from "Selective Cytotoxicity of Ascorbic Acid and Hydrogen Peroxide on Primary Human Tenon Fibroblasts without Harming Scleral Fibroblasts In Vitro" to "Selective Toxicity of Ascorbic Acid and Hydrogen Peroxide on Human Tenon Cells without Harming Scleral Cells In Vitro: A Possible Alternative to Non-Selective Mitomycin C?" This title better reflects our aim to identify safer alternatives to MMC for potential clinical use.

Unfortunately, due to ethical considerations, we have not been able to conduct in vivo studies to date. However, we hope that after the publication of this study, we will have provided sufficient evidence to justify future animal studies or off-label clinical trials.

Reviewer #1: High Passage Number for TFs: Using passage 10 Tenon fibroblasts raises concerns since higher passages might behave differently than early passage cells. A justification for choosing this passage number is required to address potential concerns about altered cell behavior.

Reviewer #2: Passage Number for TFs and SFs: The use of passage 10 TFs is concerning, as high-passage fibroblasts may differ from early-passage cells in terms of behavior. Additionally, the passage number for scleral fibroblasts (SFs) is missing and should be included in the methods for consistency and reproducibility.

Response:

The mention of passage 10 in the results section refers only to the fact that Tenon cells could be successfully expanded until passage 10 and beyond, regardless of prior intraoperative MMC treatment. This finding was included to highlight that MMC did not have a lasting inhibitory effect on cell proliferation ex vivo, which raises valid concerns about its efficacy in clinical use. However, the experiments presented in the manuscript were not conducted with passage 10 cells.

Both reviewers may have overlooked the details in the Methods section regarding the passage numbers of the Tenon fibroblasts (TFs) and Scleral fibroblasts (SFs). We want to clarify that the cells used in our experiments were between passages 2 and 8, as clearly stated in the manuscript. This passage range was chosen to ensure optimal cell behavior and reproducibility.

Moreover, we used patient-matched Tenon and Scleral fibroblasts (SFs) within similar passage ranges to ensure consistency and reproducibility. We maintained a strict protocol to use only passage-matched cells with a variation of ±1 passage number. The results presented in the manuscript are based on three technical replicates for each concentration, and the experiments were repeated three 2 to 6 times, necessitating higher passage cells due to the large number of cells required for these experiments. Only early passage cells (e.g., passage 3 or 4) would not have provided sufficient material for replications and statistical analysis.

In addition, it is difficult to define a universal "limit" for the passage number beyond which primary cells become unsuitable for in vitro studies, as this can vary greatly depending on the cell type and cell culture conditions, such as seeding density. For example, fibroblasts are more robust and can be used at higher passages than more delicate cells, such as neurons. For consistency, the passage number for SFs will also be included in the revised manuscript.

Finally, the fact that we observed consistent results across multiple independent experiments, despite using higher passage numbers, strongly suggests that the effects we report, whether with MMC, AA, or H2O2, are not passage-number dependent.

However, to avoid confusion, we revised that section stating now:" Remarkably, two out of six tenon tissue samples were treated intraoperatively with 0.04 % MMC for 3 to 5 minutes before the tissue was removed, whereas the other four tenon tissues were removed before MMC was applied. However, an outgrowth of proliferating TFs was observed in all tenon tissue samples after 5-7 days, regardless of whether they were treated intraoperatively with MMC (Fig. 1a+b, Supplementary Video S1). We could expand successfully and culture TFs up to and beyond passage 10, regardless of intraoperative MMC treatment. This observation suggests that MMC did not exert a lasting antiproliferative effect in ex vivo conditions."

Reviewer #1: Statistical Methods: The use of a t-test in comparing more than two groups is not appropriate. A better statistical approach, such as ANOVA, should be applied when comparing three groups.

Reviewer #2: Statistical Analysis Issues: In some comparisons where more than two groups are analyzed, a t-test is used, which is inappropriate. A more suitable statistical method like ANOVA should be used.

Response:

We thank the reviewer for pointing out the issue regarding the statistical methods. We acknowledge that a t-test is not the optimal choice when comparing more than two groups. For comparisons involving multiple groups, we should indeed have applied a one-way ANOVA to ensure the correct statistical treatment of the data.

In the revised manuscript, we have replaced the t-tests with ANOVA where applicable and report the results accordingly. Additionally, for post hoc analysis, we will include appropriate tests (e.g., Tukey's HSD) to identify significant differences between specific groups, if needed.

During our re-evaluation of the statistical analysis, we noted an error in the calculation of significances for the Western blot quantification in Figure 4. Upon applying the appropriate ANOVA approach, we found discrepancies in the initial significance results. We have corrected these calculations and revised Figure 4 accordingly to accurately represent the statistical outcomes.

Response to Reviewer #1

Reviewer #1: TGF-β Inclusion: The rationale for using Transforming Growth Factor-beta (TGF-β) is unclear. The authors should explain the reason for incorporating TGF-β, especially given its role in fibrosis.

Response:

We appreciate the reviewer's request for clarification. As described in the literature, TGF-β was incorporated into our experiments to demonstrate that the Tenon fibroblasts (TFs) used in the study respond to TGF-β by undergoing epithelial-mesenchymal transition (EMT). The response to TGF-β is a crucial marker of the ability of TFs to react to fibrotic stimuli. To confirm this, we conducted immunocytochemical (ICC) staining, revealing that after one week of exposure to TGF-β, the TFs show the typical expression and distribution of EMT markers such as ACTA2, fibronectin, and vimentin. These markers clearly showed that the TFs responded as expected under pro-fibrotic conditions.

We will clarify this rationale in the revised manuscript and provide more detail.

Reviewer #1: Omission in Methods: Line 241 references treatments with MMC, but this is missing from the methods section. All treatments and procedures should be clearly outlined for better transparency.

Response:

We appreciate the reviewer pointing out this omission in the methods section.

We apologize for this oversight and correct it in the revision, Stating:

“Mitomycin C (MMC) Treatment

For the intraoperative application of Mitomycin C (MMC), a concentration of 0.04% MMC was prepared and applied using a sponge. The MMC was left in contact with the tissue for three minutes to allow for adequate absorption. Following the application, the area was washed twice with 10 ml of Hanks' Balanced Salt Solution (HBSS) to remove any residual MMC.

For the co-culture experiments,…”

Reviewer #1: pH Measurement: Only the pH of the 8 mM AA solution is reported. It is important to confirm whether other concentrations maintained a stable pH, as this could affect the results.

Response:

We appreciate the reviewer's request for clarification regarding the pH of the AA solutions. The pH of the highest concentration of AA (8 mM) was measured to be

---

## [Decision Letter · Decision Letter 1]

8 Jan 2025

PONE-D-24-26851R1Selective Toxicity of Ascorbic Acid and Hydrogen Peroxide on Human Tenon Cells without Harming Scleral Cells In Vitro: A Possible Alternative to Non-Selective Mitomycin C?PLOS ONE

Dear Dr. Fuchs,

Thank you for submitting your manuscript to PLOS ONE. After careful consideration, we feel that it has merit but does not fully meet PLOS ONE’s publication criteria as it currently stands. Therefore, we invite you to submit a revised version of the manuscript that addresses the points raised during the review process.

We look forward to receiving your revised manuscript.

Kind regards,

Sanaz Alaeejahromi

Academic Editor

PLOS ONE

**Journal Requirements:**

Reviewers' comments:

Reviewer's Responses to Questions

**Comments to the Author**

1. If the authors have adequately addressed your comments raised in a previous round of review and you feel that this manuscript is now acceptable for publication, you may indicate that here to bypass the “Comments to the Author” section, enter your conflict of interest statement in the “Confidential to Editor” section, and submit your "Accept" recommendation.

Reviewer #3: All comments have been addressed

2. Is the manuscript technically sound, and do the data support the conclusions?

Reviewer #3: Yes

3. Has the statistical analysis been performed appropriately and rigorously? 

Reviewer #3: Yes

4. Have the authors made all data underlying the findings in their manuscript fully available?

Reviewer #3: Yes

5. Is the manuscript presented in an intelligible fashion and written in standard English?

Reviewer #3: Yes

6. Review Comments to the Author

**Reviewer #3: ** (No Response)

7. PLOS authors have the option to publish the peer review history of their article (what does this mean? ). If published, this will include your full peer review and any attached files.

**Do you want your identity to be public for this peer review?** For information about this choice, including consent withdrawal, please see our Privacy Policy .

Reviewer #3: No

---

## [Author Response · Author response to Decision Letter 2]

18 Feb 2025

Dear Reviewers,

We sincerely appreciate the opportunity to submit a revised version of our manuscript titled "Selective Toxicity of Ascorbic Acid and Hydrogen Peroxide on Human Tenon Cells without Harming Scleral Cells In Vitro: A Possible Alternative to Non-Selective Mitomycin C?" (Manuscript ID: PONE-D-24-26851R1). We thank you and the reviewers for your constructive feedback, which has helped us improve the quality of our work.

We are pleased to note that Reviewer #3 has confirmed that all comments raised in the previous round of review have been adequately addressed. Moreover, the reviewer has affirmed that the manuscript is technically sound, the statistical analyses have been conducted appropriately, and the data fully support our conclusions. Additionally, the reviewer confirmed that all data are fully available and that the manuscript is well-written and intelligible.

We also sincerely appreciate the reviewer’s detailed corrections and suggestions within the manuscript. We have carefully reviewed and incorporated these proposed changes to further enhance clarity and accuracy.

As no further specific concerns or revision requests have been raised by the reviewers, we have carefully reviewed the manuscript to ensure clarity, correctness, and compliance with the journal’s formatting and reference requirements. We have also verified that all cited references are accurate and up to date, in line with the editorial guidelines.

Accordingly, we are resubmitting the revised manuscript along with the required files:

- A marked-up copy of the manuscript highlighting all changes made.

- A clean version of the revised manuscript without tracked changes.

- This rebuttal letter summarizing our response to the review process.

We appreciate the time and effort invested by the reviewers and the editorial team in assessing our manuscript, and we hope that the revised version meets all publication criteria. Please let us know if any further modifications are required.

The authors confirm that this manuscript has not been previously published and is not currently being considered by any other journal. We have not had any prior discussions with a PLOS ONE Board Member regarding the work described in the manuscript. All authors have approved the contents of this paper and have adhered to PLOS ONE's submission policies.

Thank you again for your consideration. We look forward to your positive response.

Best regards,

Heiko Fuchs

---

## [Editor Report · Decision Letter 2]

21 Feb 2025

Selective Toxicity of Ascorbic Acid and Hydrogen Peroxide on Human Tenon Cells without Harming Scleral Cells In Vitro: A Possible Alternative to Non-Selective Mitomycin C?

PONE-D-24-26851R2

Dear Dr. Heiko Fuchs

We’re pleased to inform you that your manuscript has been judged scientifically suitable for publication and will be formally accepted for publication once it meets all outstanding technical requirements.

Kind regards,

Sanaz Alaeejahromi

Academic Editor

PLOS ONE

---

## [Editor Report · Acceptance letter]

PONE-D-24-26851R2

PLOS ONE

Dear Dr. Fuchs,

I'm pleased to inform you that your manuscript has been deemed suitable for publication in PLOS ONE. Congratulations! Your manuscript is now being handed over to our production team.

Kind regards,

on behalf of

Dr. Sanaz Alaeejahromi

Academic Editor

PLOS ONE